# Genome-Wide Identification and Characterization of Auxin Response Factor (ARF) Gene Family Involved in Wood Formation and Response to Exogenous Hormone Treatment in *Populus trichocarpa*

**DOI:** 10.3390/ijms24010740

**Published:** 2023-01-01

**Authors:** Yingying Liu, Ruiqi Wang, Jiajie Yu, Shan Huang, Yang Zhang, Hairong Wei, Zhigang Wei

**Affiliations:** 1State Key Laboratory of Tree Genetics and Breeding, Northeast Forestry University, Harbin 150040, China; 2Engineering Research Center of Agricultural Microbiology Technology, Ministry of Education, Heilongjiang University, Harbin 150080, China; 3Heilongjiang Provincial Key Laboratory of Plant Genetic Engineering and Biological Fermentation Engineering for Cold Region, School of Life Sciences, Heilongjiang University, Harbin 150080, China; 4College of Forest Resources and Environmental Science, Michigan Technological University, Houghton, MI 49931, USA

**Keywords:** genome-wide analysis, auxin response factor (ARF), *Populus trichocarpa*, wood formation, exogenous hormone

## Abstract

Auxin is a key regulator that virtually controls almost every aspect of plant growth and development throughout its life cycle. As the major components of auxin signaling, auxin response factors (ARFs) play crucial roles in various processes of plant growth and development. In this study, a total of 35 *PtrARF* genes were identified, and their phylogenetic relationships, chromosomal locations, synteny relationships, exon/intron structures, *cis*-elements, conserved motifs, and protein characteristics were systemically investigated. We also analyzed the expression patterns of these *PtrARF* genes and revealed that 16 of them, including *PtrARF1*, *3*, *7*, *11*, *13*–*17*, *21*, *23*, *26*, *27*, *29*, *31*, and *33*, were preferentially expressed in primary stems, while 15 of them, including *PtrARF2*, *4*, *6*, *9*, *10*, *12*, *18*–*20*, *22*, *24*, *25*, *28*, *32*, and *35*, participated in different phases of wood formation. In addition, some *PtrARF* genes, with at least one *cis*-element related to indole-3-acetic acid (IAA) or abscisic acid (ABA) response, responded differently to exogenous IAA and ABA treatment, respectively. Three PtrARF proteins, namely PtrARF18, PtrARF23, and PtrARF29, selected from three classes, were characterized, and only PtrARF18 was a transcriptional self-activator localized in the nucleus. Moreover, Y2H and bimolecular fluorescence complementation (BiFC) assay demonstrated that PtrARF23 interacted with PtrIAA10 and PtrIAA28 in the nucleus, while PtrARF29 interacted with PtrIAA28 in the nucleus. Our results provided comprehensive information regarding the *PtrARF* gene family, which will lay some foundation for future research about *PtrARF* genes in tree development and growth, especially the wood formation, in response to cellular signaling and environmental cues.

## 1. Introduction

Indole-3-acetic acid (IAA), the primary type of auxin, plays crucial roles during plant growth and development in response to cellular signaling and environmental cues [1,2]. What is certain is that most of these functions are achieved through regulating gene expressions via auxin response factor (ARF) proteins, which translate the IAA signal into gene expressions with the help of auxin/indole-3-acetic acid (Aux/IAA) proteins [3]. Most ARF proteins generally consist of three conserved domains [4]: (1) an N-terminal B3-type DNA binding domain (DBD) responsible for binding to auxin response elements (AREs); (2) a variable middle region (MR) that functions as an activation domain (AD) or repression domain (RD) responsible for causing ARFs to be activators or repressors [5,6]; and (3) a carboxy-terminal dimerization domain (CTD) responsible for protein–protein interactions, such as homodimerization of ARFs or the heterodimerization of ARFs and Aux/IAAs [7]. It is noteworthy that some ARFs only have parts of these three conserved domains. For example, ARF3, ARF13, and ARF17 lack CTD, and ARF23 only consists of a truncated DBD in *Arabidopsis* [8].

With increasing numbers of sequenced genomes in plants, *ARF* gene families have been studied at whole-genome level in *Oryza sativa* [9], *Citrus sinensis* [10], *Populus trichocarpa* [11], and many other plant species [12,13,14]. These previous studies demonstrated that the structures of proteins encoded by *ARF* genes differentiated greatly within family or families among these species. For example, compared with *Arabidopsis thaliana* [15], only one ARF with a truncated DBD was found in *Citrus sinensis* [10], whereas a large number of ARFs lacking CTD have been found in *Oryza sativa* [9], *Zea mays* [16], *Musa acuminata* [17], and *Medicago truncatula* [12]. Thus, the *ARF* gene families in different plant species exhibit a particularly high diversity and abundance, most of which require further studies and characterization for a better understanding.

The accumulated evidence demonstrated that *ARF* genes involved multiple auxin-dependent processes in plants [18,19,20]. For example, *AtARF2–4* and *AtARF5* are essential for female and male gametophyte development, respectively [21]. *AtARF8* regulates stamen elongation and endothecium lignification [22], while *AtARF3* plays a distinct role during early flower development [23]. *AtARF7* and *AtARF19* are essential for auxin-mediated plant development by regulating both unique and partially overlapping sets of target genes [15]. In addition, only five AtARF proteins (AtARF5, AtARF6, AtARF7, AtARF8, and AtARF19) have been characterized as transcriptional activators, while the other AtARF proteins were classified as repressors [4,24]. Moreover, the expressions of many *ARF* genes were altered in responsive to various abiotic stresses, such as drought [25,26], salt [17,27,28], and cold [10,17,29,30,31]. These studies provided fundamental information for the subsequent identification of each *ARF* gene function during plant growth and development in response to cellular signaling and environmental stress.

*P. trichocarpa* has been long recognized as a woody model plant species enabling detailed and integrated analysis of complicated traits of secondary growth (wood formation), perennial growth (flowering, bud dormancy, and break), and adaptive traits (adaptive strategies in defending multiple stresses), which are not fully developed in annual herbaceous species [32] such as *A. thaliana* and *O. sativa* [33,34]. Due to their importance in tree growth and development, the *PtrARF* gene family was identified from *P. trichocarpa* genome v1.1 release [35] and analyzed in 2007 [11]. However, the early genome assembly using shotgun technology and gene annotation with less-efficient tools led to wrongly annotated gene models. With the continuous release of the new versions of P. trichocarpa genome, many gene models in v1.1 annotation have been abolished. For example, the *PtrARF* ID numbers set in previous study, such as estExt_fgenesh4_pg.C_1500013, estExt_fgenesh4_pm.C_LG_XII0386, and estExt_Genewise1_v1.C_LG_IV2935, have been abandoned in the *P. trichocarpa* v3.0, which caused the previous knowledge derived from version 1.1 not to be easily utilized in present *PtrARF* gene study. This kind of problem has been also demonstrated by the research on the *R2R3-MYB* gene family [36]. In addition, as exponentially increasing amounts of RNA-Seq data for the poplar are available [37], it is greatly helpful to update and enrich the knowledge of the *PtrARF* gene family at the whole-genome level.

In the present study, based on *P. trichocarpa* v3.0, we identified the whole *PtrARF* gene family, which comprised 35 *PtrARF* genes, and performed systematic analysis of the basic characteristics of the poplar *ARF* family at the whole-genome level. The expression patterns of some *PtrARF* genes were analyzed during wood formation and under exogenous IAA and ABA treatment. In addition, we investigated the transcriptional activity properties and subcellular locations of PtrARF18, PtrARF23, and PtrARF29 and their interactions with PtrIAA10 and PtrIAA28 through yeast two-hybrid (Y2H) and bimolecular fluorescence complementation (BiFC), respectively. Our results lay an important foundation for future studies aimed at understanding the functions of *PtrARF* genes in wood formation and abiotic stresses responses.

## 2. Results

### 2.1. Identification and Phylogeny of PtrARF Genes

Through ortholog blasting search against *P. trichocarpa* v3.0 in Phytozome12 using known ARF amino acid sequences of *A. thaliana*, a total of 35 protein sequences were confirmed as ARF family members of poplar (*PtrARF*). Subsequently, the 35 *PtrARF* genes were successively designated as *PtrARF1* to *PtrARF35* according to the order of the homologous chromosomes (Appendix A). Then, in order to clearly understand the characteristics of *PtrARF* gene, the basic information, such as gene name, protein length, predicted isoelectric point (pI), molecular weight (MW), and the typic conserved domain, was analyzed and is shown in Appendix A, which demonstrates that *PtrARF* proteins generally have 335–1141 amino acids, pI from 5.25 to 8.18, and MW from 37.729 to 127.603 kDa, indicating that PtrARF proteins could function in diverse microenvironments.

To investigate the grouping pattern and genetic relationships of 35 PtrARF genes, the typical conserved domain (PF02362) sequences of *P. trichocarpa* (35), *A. thaliana* (23), *E. grandis* (17), and *O. sativa* (25) were subjected to phylogenetic analysis. As shown in Figure 1, these 100 ARF proteins were divided into three classes, corresponding to three groups of AtARF proteins as defined by Okushima in *Arabidopsis* [15].

The *ARF* genes of *P. trichocarpa*, *A. thaliana*, *E. grandis*, and *O. sativa* could be assembled in the same evolutionary class (Figure 1), demonstrating that the plant *ARF* gene families were relatively conservative among different species, and *ARF* gene evolution was later than that of herbs and woody plant and monocotyledons and dicotyledons, respectively. It was interesting that *PtrARF* genes were first grouped with *AtARF* genes but not *EgrARF* genes in most cases, which indicated that *PtrARF* genes might have a closer evolutionary relationship with *AtARF* genes than with *EgrARF* genes. However, a small part of *PtrARF* genes, such as *PtrARF28* together with *OsARF* genes such as *OsARF23*, were clustered into same classes, indicating that the evolutionary relationship of some *ARF* genes between poplar and rice were more conserved than that between monocots and dicots.

As shown in Figure 1, class I was the largest class, which was further divided into three subclasses: class Ia, including 11 *PtrARF* genes; class Ib, only containing one gene, *PtrARF29*; and class Ic, comprising three *PtrARF* genes, whereas class II and class III contained 12 and 8 *PtrARF* members, respectively. In addition, *PtrARF* genes were almost interspersed and distributed in each class, suggesting that the *PtrARF* genes arose before the lineage spilt, and the evolutionary relationship of *PtrARF* genes is relatively complex. Furthermore, the phylogenetic tree shows that there were 13 sister pairs among 35 *PtrARF* genes, and each class consisted of one or more sister gene pairs (Figure 1), suggesting the duplication events played a major role in the expansion of the *PtrARF* gene family.

### 2.2. Chromosome Distribution and Synteny Relationship

#### 2.2.1. Chromosome Distribution

As shown in Figure 2, 35 *PtrARF* genes were unevenly anchored to 16 of the total 19 chromosomes of poplar, with no *PtrARF* gene distribution in chromosome 7, 13, and 19 or scaffolds. Chromosome 2 contained the largest number (five) of *PtrARF* genes, whereas Chr1, as the largest chromosome, only has three *PtrARF* genes, and most of the *PtrARF* genes were evenly distributed on other 15 chromosomes, suggesting the number of *PtrARF* genes on each chromosome was irrelevant to the chromosome size. Notably, collinearity analysis of the *PtrARF* genes showed nine pairs of homologous *PtrARF* genes, which illustrated that these nine *PtrARF* pairs were formed by segmental duplication (Appendix A). However, no tandem duplication was identified among the remaining four *PtrARF* gene pairs, as these genes were separated by at least several megabases, indicating these four gene pairs might form through whole-genome duplication but not tandem duplication. In addition, 35 *PtrARF* genes almost evenly anchored on positive and reverse strands of the chromosome, respectively, with 18 *PtrARF* genes, namely *PtrARF3*–*7*, *9*–*11*, *13*, *19*–*22*, *27*, *29*, *30*, *34*, and *35*, located on the positive chromosomes and the other remaining 17 *PtrARF* genes located on the reverse chromosome strands (Appendix A). Altogether, these results suggest that expansion of *PtrARF* gene family mainly originated from segmental duplication, followed by whole-genome duplication, while tandem duplication was not involved during the *PtrARF* gene family expansion.

#### 2.2.2. Synteny Relationship of *PtrARF* Genes

To examine the evolutionary origins and orthologous relationships of *ARF* genes among diverse species, the synteny analysis of three dicotyledons (*P. trichocarpa*, *E. grandis*, and *A. thaliana*) and one monocotyledon (*O. sativa*) was performed. As shown in Figure 3, there were 3 pairs of *ARF* orthologous genes between *A. thaliana* and *P. trichocarpa* and 12 pairs between *E. grandis* and *P. trichocarpa*, suggesting during species divergence; 2 and 12 *P. trichocarpa ARF* genes share common ancestors with *A. thaliana* and *E. grandis*, respectively. In addition, two genes (*PtrARF2* and *PtrARF10*) were discovered among *A. thaliana*, *P. trichocarpa*, and *E. grandis*, indicating that these two genes were possibly inherited from an even earlier land plant ancestor. In contrary, no synteny was identified in *P. trichocarpa* vs. *O. sativa*, suggesting that multiple *ARF* genes may be formed after the differentiation of monocotyledon and dicotyledon.

#### 2.2.3. K Values of Homologous and Orthologous *PtrARF* Gene Pairs

Ka/Ks ratio is usually an important indicator of selective pressure in evolution [38]. The results of homologous and orthologous *PtrARF* gene pairs analysis revealed that among nine sister gene pairs originated from segment duplication, eight paralogous pairs (*PtrARF2/10*, *PtrARF4/18*, *PtrARF5/17*, *PtrARF6/16*, *PtrARF9/15*, *PtrARF19/35*, *PtrARF22/25*, and *PtrARF28/31*) showed a Ka/Ks ratio less than 0.3, suggesting their functional conservation, whereas only one paralogous pair (*PtrARF20/32*) had a Ka/Ks ratio more than 0.3, indicating the possibility of significant functional divergence after duplication. In addition, the Ka/Ks ratio of segmental duplication ranged from 0.16 to 0.31, with an average value of 0.22, indicating that the *PtrARF* gene family has undergone strong purification selection after duplication (Appendix A.). Furthermore, 15 orthologous *ARF* gene pairs between poplar and other two species showed a Ka/Ks ratio less than 0.3 (Appendix A), demonstrating their function relative conservation between poplar and *Arabidopsis* and *Eucalyptus*, respectively.

### 2.3. Gene Structures and Conserved Domains

The intron/exon analysis showed that the numbers of introns and exons of PtrARF genes ranged from 1 to 14 and 2 to 15 across distinct classes, respectively (Figure 4B and Appendix A). The PtrARF genes in class I contained 8–15 exons and 7–14 introns, class II contained 13–14 exons and 12–13 introns, and class III had 2–4 exons and 1–3 introns, suggesting that the intron/exon structures were somewhat differentiated over different classes, whereas they were conserved among the PtrARF genes within subclasses.

To further explore the functional diversity of PtrARF proteins, motifs within these proteins were investigated. As shown in Figure 4C and Appendix A, a total of 20 conserved motifs were identified in the three classes, and each class contained diverse numbers of motifs. For example, class I and class III possessed 15 conserved motifs, while class II possessed 18 conserved motifs. In addition, the three classes had 11 common motifs, namely motifs 1–6, 8–10, 17, and 18; while the class I contained four specific motifs, i.e., motifs 7, 11, 12, and 19; and the motifs 13–15 and 20 were specific to class III. We found that the BD of PtrARF proteins comprised motifs 1 and 2; motifs 4, 6, and 8 corresponded to MR; and motifs 7 and 9 corresponded to CTD (Appendix A).

Meanwhile, as shown in Figure 5, according to the presence or absence of CTDs and MR amino-acid compositions, the 35 PtrARF proteins were divided into three groups: (1) PtrARFs, with a DBD, activator MR (enriched with glutamine), and a CTD, such as PtrARF3–5, 13, 17–19, 21, 27, and 33–35; (2) PtrARFs, with a DBD, repressor MR (enriched with serine, leucine, proline, and glycine), and a CTD, including PtrARF1, 2, 7, 9–11, 15, 23, and 28–30; and (3) PtrARFs, with a DBD and repressor MR but no CTD, containing of PtrARF6, 8, 12, 14, 16, 20, 22, 24–26, 31, and 32. In addition, the PtrARF activators and repressors showed uneven distribution among three classes, with class I including both activators and repressors, class II only containing activators, and class III only comprising repressors.

Altogether, the results not only suggest that *PtrARF* genes of different classes might evolve different functions, but they also supported the reliability of *PtrARF* genes classification.

### 2.4. Cis-Elements in the Promoters of PtrARF Genes

The expression of a gene involved in a specific biological process is modulated by upstream transcription factors through binding to its promoter specific *cis*-elements. Thus, in order to explore the biological functions of *PtrARF* genes, the *cis*-elements within 3 kb upstream sequences of each *PtrARF* gene were analyzed by *PlantCARE Database*. As shown in Figure 6 and Appendix A, 35 *PtrARF* genes contained a total of 106 different *cis*-elements, which mainly contained: (1) *cis*-elements responding to hormones, including IAA, ABA, gibberellin acid (GA), ethylene, and methyl jasmonic acid; (2) *cis*-elements related to stress involved in drought, low temperature, and heat; (3) *cis*-elements involved in development and tissue specificity; (4) *cis*-elements related to circadian control; and (5) *cis*-elements related to xylem development, such as SNBE [39], M46RE [40], ACI, ACII, ACIII [41], and SMRE [42] corresponding to secondary cell wall (SCW) formation and TERE [43] involved in programmed cell death (PCD) (Appendix A). Overall, the diversity of *cis*-elements implied that *PtrARF* genes participated in many biological processes and could be linkers between multiple hormone response and other important biological processes.

The numbers of *cis*-elements exhibited obvious differences among the three classes (Appendix A). For example, there were 3427, 2696, and 1772 *cis*-elements in the class I, II, and III, respectively. In addition, although the three classes contained 67 common *cis*-elements, each class contained unique *cis*-elements (Appendix A). For example, class I contained six unique *cis*-elements such as CAG-motif, light responsive element, and MSA-like element involved in cell cycle regulation; class II included four unique *cis*-elements such as chs-Unit 1 m1, AC-I, and Gap-box; and class III comprised four unique *cis*-elements such as GTGGC-motif, L-box, and Box III. These results further suggest that *PtrARF* genes of different classes could evolve somewhat diverse functions during evolution.

### 2.5. Expression Characteristics of PtrARF Genes

Considering the importance of secondary growth and adaptation for perennial trees, we focused on the analysis of putative functions of *PtrARF* genes in wood formation and abiotic stress. Since the ABA is known as the “stress hormone” related to multiple abiotic stresses [17,25,31], we analyzed the expression patterns of *PtrARF* genes under exogenous ABA treatment instead of an abiotic stress. In addition, the different zones along the vertical stems of poplar represent the different developmental phases of wood formation [44] and thus were used to investigate the roles of *PtrARF* genes in different stages of wood formation. Furthermore, though *ARF* genes are known as the core components of the typic transcriptional response pathway of IAA [3], the contribution of each *PtrARF* gene in this process was still unclear. Therefore, the expression pattern of each *PtrARF* gene also needs to be determined in response to the exogenous IAA treatment.

#### 2.5.1. Expression Patterns of PtrARF Genes in Wood Formation

Based on expression profiles of different developmental stems in poplar [44], we found that 15 *PtrARF* genes, among which five, four, and six *PtrARF* genes belonged to classes I, II, and III, respectively, exhibited preferable expression in secondary xylem tissues among three stem stages (Figure 7A and Appendix A). In addition, 16 *PtrARF* genes, among which eight, six, and two *PtrARF* genes belonged to classes I, II, and III, respectively, were preferably expressed in primary xylem tissues (Figure 7A and Appendix A). To verify this, we further detected the expression levels of these *PtrARF* genes in diverse development stems of poplar by qRT-PCR analysis. As shown in Figure 7B, the expression patterns of these *PtrARF* genes in the stems of different development stages were highly consistent with those of previous expression profiles of these genes [44], suggesting it is reasonable to use RNA-seq data to assess the expression levels of *PtrARF* genes.

To reveal the exact functions of these 15 *PtrARF* genes related to different processes of wood formation, we further investigated their expression levels based on the AspWood RNA-seq dataset [45]. The results showed that these 15 *PtrARF* genes exhibited different expression levels during diverse phases of wood formation, suggesting that these *PtrARF* genes played distinct roles during wood formation (Figure 7C and Appendix A). For example, *PtrARF4*, *PtrARF18*, and *PtrARF35* exhibited the preferential high expression levels in cambium differentiation and xylem expansion, suggesting that these genes played important roles in regulating early development processes of wood formation. Seven *PtrARF* genes, namely *PtrARF9*, *PtrARF10*, *PtrARF12*, *PtrARF20*, *PtrARF22*, *PtrARF25*, and *PtrARF28*, had higher expression levels during the SCW formation, implying that these genes participated in regulating cellulose, hemicellulose, and lignin biosynthesis. Nine *PtrARF* genes, namely *PtrARF2*, *PtrARF6*, *PtrARF9*, *PtrARF10*, *PtrARF20*, *PtrARF22*, *PtrARF24*, *PtrARF25*, and *PtrARF32*, showed the highest expression levels in xylem maturation, suggesting that these *PtrARF* genes mainly functioned in lignification and PCD process of wood formation. Altogether, these results indicate that *PtrARF* genes evolved diverse biological functions that are important for wood formation.

#### 2.5.2. Expression Patterns of PtrARF Genes in Response to the Exogenous ABA and IAA Treatments

The expression levels of 15 *PtrARF* genes, which were selected due to the presence of both ABA and abiotic stress response *cis*-elements in their promoter regions, exhibited obvious oscillatory changes in roots, stems, and leaves under exogenous ABA treatment (Figure 8A–C). In addition, the expression levels of these 15 *PtrARF* genes peaked at different time points among three poplar organs. For example, most of the 15 *PtrARF* genes showed the highest expression levels at 6 h in roots, 24 h in stems, and 6 h and 12 h in leaves under exogenous ABA treatment. These results demonstrated that 15 *PtrARF* genes played different roles in different organs of poplar in response to abiotic stress and ABA signaling.

Under exogenous IAA treatment, 12 *PtrARF* genes, containing at least one *cis*-element related to IAA response, showed differential expression patterns among roots, stems, and leaves (Figure 8D–F). Of these genes, seven *PtrARF* genes in roots had the highest expression levels at 72 h compared with other timepoints under exogenous IAA treatment. In stems, seven *PtrARF* genes had the highest expression level at 48 h and five *PtrARF* genes at 12 h. In leaves, there were two time points for most of the 12 *PtrARF* genes, with the higher expression levels under exogenous IAA treatment. For example, five *PtrARF* genes had the higher expression level at 72 h and six *PtrARF* genes at 24 h under exogenous IAA treatment. These results suggest that *PtrARF* genes have different roles in IAA signaling among different organs of poplar.

### 2.6. Transcriptional Properties of PtrARF Proteins

#### 2.6.1. Subcellular Localizations of PtrARF Proteins

PtrARF18, PtrARF23, and PtrARF29 were selected from three classes. PtrARF18, with glutamine enriched in its MR, was considered as transcriptional activator, while PtrARF23 and PtrARF29, with serine, leucine, proline, and glycine enriched in their MR, were believed as transcriptional repressors. Except these two criteria, they are randomly selected targets.

To confirm the subcellular locations of PtrARF proteins predicted by Plant-mPLoc, PtrARF18, PtrARF23, and PtrARF29 were selected from three classes to perform subcellular localization analysis. In addition, based on the protein–protein interactions between AtARFs and AtIAAs predicted by STRING, two orthologs of AtIAA proteins in poplar, i.e., PtrIAA10 and PtrIAA28, were hypothesized to interact with these three selected corresponding PtrARF proteins. Given this, PtrIAA10 (*Potri.002G256600*) and PtrIAA28 (*Potri.003G048100*) were also selected to subcellular localization analysis. As shown in Figure 9A, the green fluorescent protein (GFP) signals of the positive controls were observed throughout the protoplasts, whereas fluorescent signals of five fusion proteins were only found in the nuclei, suggesting that the three PtrARF and two PtrIAA proteins were nucleoproteins.

#### 2.6.2. Transcriptional Activity of PtrARF Proteins

PtrARF18 was considered as transcriptional activator, while PtrARF23 and PtrARF29 were believed to be transcriptional repressors. To verify this, PtrARF18, PtrARF23, and PtrARF29 were fused with the GAL4 DNA-binding domain, respectively, and we tested their potential to activate the reporter gene expression in yeast. As shown in Figure 9B, only PtrARF18 could activate the expression of His3, Ade2, and Mel1 reporter genes, suggesting PtrARF18 was a transcriptional self-activator, whereas the PtrARF23 and PtrARF29 had no transcriptional self-activation ability.

### 2.7. Interactions between PtrARF23/29 and PtrIAA10/28

As a transcriptional self-activator, PtrARF18 was not viable fort testing to determine whether it interacted with other proteins according to Y2H-Gold system assay. Therefore, only two PtrARF repressors, namely PtrARF23 and PtrARF29, were used to verify their putative interactions with PtrIAA10 and PtrIAA28 using Y2H and bimolecular fluorescence complementation (BiFC) assays.

As shown in Figure 9C, the results of Y2H assays showed that PtrARF23-BD and PtrIAA28-AD, PtrARF23-BD and PtrIAA10-AD, and PtrARF29-BD and PtrIAA28-AD, except for PtrARF29-BD and PtrIAA10-AD, exhibited blue on SD/-Leu/-Trp/-His/-Ade/x-α-gal, confirming the interaction specificity between PtrARF23 and PtrIAA10/28 and PtrARF29 and PtrIAA28.

As shown in Figure 9D, no yellow fluorescent protein (YFP) signal was detected when PtrARF23 or PtrARF29-nYFP was co-expressed with cYFP or PtrIAA10 or PtrIAA28-cYFP was co-expressed with nYFP. In contrary, co-expression of PtrARF23 and PtrARF29 fused to the amino-terminal half of YFP (nYFP) and PtrIAA10 and PtrIAA28 fused to the carboxy-terminal half (cYFP) of yellow florescent protein led to visible fluorescence in the nucleus of co-transformed protoplasts (Figure 9D). These results further confirmed that there were specific interactions between PtrARF23/29 and PtrIAA10/28.

## 3. Discussion

Since auxin plays a pivotal role in the regulation of plant growth and development, *ARF* genes are key components of plant auxin signaling during these processes [3] and thus are critically important in forest trees that are subjected to abiotic stress and secondary growth, especially vascular cambium-based growth. Genomic resources are the starting point for uncovering gene regulation mechanisms of the unique traits present in a given plant species [46].

In a previous study [11], a total of 39 predicted ARF genes were found in the *P. trichocarpa* v1.1 database, but conserved domain evaluations showed that four gene models (*PoptrARF3.3*, *6.3*, *16.5*, and *16.6*) lack one or more domains. In this study, 35 *ARF* genes were identified based on *P. trichocarpa* v3.0 annotation, and we confirmed the presence of the conserved B3 and Auxin_resp domains, and any redundant sequences were manually removed.

The previous *P. trichocarpa* ARF family genes study was published in 2007, when the *P. trichocarpa* genome of version 1.1 had just been released. After many years, the *P*. *trichocarpa* genome has been reassembled two times. Compared to the more recent version 3.0, a multitude of gene models in version 1.1 have been abolished, and therefore, a new study of the PtrARF family is needed. Our analysis was based on version 3.0, using the latest bioinformatics analysis methods in characterization of *PtrARF* genes in *P*. *trichocarpa* at the whole-genome level. Our study provided more accurate information for further identifying their functions in poplar growth and development, especially in wood formation and response to abiotic stress.

### 3.1. Expansion and Evolution of PtrARF Gene Family

#### 3.1.1. The Increasing Numbers and Phylogenetic Relationships of *PtrARF* Gene Family

The members of *ARF* gene family manifested a clear tendency to enlarge [47]. In this study, we identified 35 *ARF* genes from *P. trichocarpa* v3.0 by genome-wide analysis, suggesting that the *PtrARF* gene family was expanded compared to *Arabidopsis* (23) [15], *O. sativa* (25) [9], and *E. grandis* (17) [48]. Phylogeny and synteny analysis revealed that both the segmental duplications and whole-genome duplications contributed to *PtrARF* gene family expansion. However, it was somewhat inconsistent with the previous conclusion that segmental duplication was the major cause for the *ARF* gene family expansion [49,50,51].

To infer the functions of *PtrARF* genes based on the functional classes previously described in *Arabidopsis* [52], we divided 100 *ARF* genes of four species into three classes according to phylogenetic tree of *AtARF* genes [15], which was different with the four phylogenetic classes of *ARF* genes in *O. sativa* [9], soybean [53], *E. grandis* [48], tomato [54], and *B. rapa* [49]. Since many *AtARFs* and *OsARFs* have been already identified in previous reports [9,15], the phylogenetic relationships of *ARF* genes in these species could provide important information for speculating the putative biological functions of the *PtrARF* genes.

#### 3.1.2. The Diversities of PtrARF Proteins

In present study, we found that the ratio of activator (12)/repressor (23) of PtrARF proteins was 0.52, which is higher than that in *M. truncatula* (0.26) [12], *Capsicum annuum* (0.22) [13], and *Dimocarpus longan* (0.35) [55] but lower than that in *A. thaliana* (0.59) and *O. sativa* (0.56) [9,15]. However, the mechanism of the different ratio of activator/repressor of ARF proteins among plant species is still unclear [55]. In addition, the uneven distribution of activator and repressor of PtrARF proteins among three classes demonstrated the PtrARF proteins evolved distinct functions, which was also observed in other plant ARF proteins such as *Brassica napus* [50].

In addition, the percentage of CTD-truncated PtrARF (37.5%) was higher than that in *Vitis vinifera* (10.5%) [14], *A. thaliana* (17.39%) [15], *O. sativa* (24%) [9], *Brassica rapa* (22.6%) [51], and *Solanum lycopersicum* (28.6%) [56] but lower than that in *Hordeum vulgare* (45.0%) and *M. truncatula* (60.9%) [12,57]. It has been reported that the truncated ARF proteins without CTD function, whose activities were regulated through interaction with other transcription factors instead of interaction with Aux/IAA, are likely to be insensitive to auxin [4,8]. Thus, we speculated that 12 truncated PtrARF proteins could participate in plant development and growth in an auxin-independent way through interaction with other transcription factors but not PtrARF proteins. Nevertheless, the differences of the percentages of CTD-truncated ARF proteins among different plant species have been not well-understood, and this point remains open.

### 3.2. Expression Patterns of PtrARF Genes in Wood Formation and Exogenous ABA and IAA Treatments

In present study, we thoroughly analyzed the *cis*-elements of each *PtrARF* gene promoter, and the results not only revealed the regulatory molecular mechanisms of the *ARF* genes involved in plant growth and development in response to cellular signaling and environmental cues, as described in previous studies [1,2], but also provided clues for identifying the biological functions of *PtrARF* genes for a specific biological process. For example, *cis*-element analysis revealed that there were 15 *PtrARF* genes containing at least one *cis*-element related to xylem development (Figure 6), including SNBE [39], M46RE [40], ACI, ACII, ACIII [41], and SMRE [42], which play key roles in the regulation of SCW formation, and TERE [43], which is involved in PCD. In addition, based on our previous expression profiles [44] and AspWood RNA-seq dataset and qRT-PCR assay [45], we unveiled that these 15 *PtrARF* genes are involved in diverse developmental phases of wood formation (Figure 7). Similarly, about half the number of the *EgrARF* genes, including three predicted repressors (EgrARF3, 4, and 9A) and one predicted activator (EgrARF6A), were found to be down-regulated in tension wood as compared to the control upright xylem [48]. In addition, we found that 15 *PtrARF* genes might participate in response to abiotic stress and ABA treatment based on the presence of multiple *cis*-elements related to abiotic stress and ABA response in their promoter regions and their response to the exogenous ABA treatment (Figure 8). Furthermore, 12 *PtrARF* genes whose promoters contain at least one *cis*-element related to response to IAA responded to the exogenous IAA treatment. Our observations are similar to the phenomena of *PtrARF* genes in response to exogenous ABA and IAA treatments observed in other plant *ARF* genes [58,59].

### 3.3. PtrARF Proteins Interaction with PtrIAA Proteins

The Y2H and BiFC assay proved the interaction specificity between the PtrARF23 and PtrIAA10/28 and PtrARF29 and PtrIAA28, which is helpful to uncover the biologic functions of these two PtrARF proteins pairs participating in the process related to IAA signaling. Although it is not yet possible to report the exact functions of the these PtrARF proteins and their products, our results obtained in the present study formed a valuable basis for future research on the biological functions of PtrARF proteins in the regulatory mechanisms of auxin signaling, wood formation, and abiotic stress in poplar.

## 4. Materials and Methods

### 4.1. Identification of ARF Gene Family and Their Chromosome Distributions in P. trichocarpa Genome

To define members of *ARF* gene family in *P. trichocarpa*, amino acid sequences of 23 known *ARF* genes in *Arabidopsis* (*AtARFs*) were obtained from TAIR (https://www.arabidopsis.org/ (accessed on 15 February 2021)). These *Arabidopsis* ARF protein sequences were used as a query to blast *P. trichocarpa* v3.0 in Phytozome 12 (https://phytozome-next.jgi.doe.gov/blast-search (accessed on 15 February 2021)). The BLASTP search results were further examined using the online tools SMART (http://smart.embl-heidelberg.de/smart/set_mode.cgi?GENOMIC=1 (accessed on 16 February 2021)) and Pfam (https://pfam-legacy.xfam.org/ (accessed on 16 February 2021)) to confirm the presence of the conserved B3 (PF02362) and Auxin_resp (PF06507) domains, and any redundant sequences were manually removed.

The genome, transcript, CDS, peptide, and 3000 bp upstream of the translational start site (ATG) promoter region sequences were obtained from *P. trichocarpa* v3.0 in Phytozome 12 (https://phytozome-next.jgi.doe.gov/info/Ptrichocarpa_v3_0 (accessed on 18 February 2021)). The positions of *PtrARF* genes and chromosome size were obtained from *P. trichocarpa* v3.0 and visualized by the TBtools [60]. The *cis*-elements of promoters were predicted using PlantCARE (http://bioinformatics.psb.ugent.be/webtools/plantcare/html/ (accessed on 18 February 2021)). The Venn diagrams of *cis*-elements in each class were drawn by Venny 2.1 (https://bioinfogp.cnb.csic.es/tools/venny/index.html (accessed on 20 February 2021)). Furthermore, the *cis*-elements related to hormone response and xylem development were mapped by the IBS web server (http://ibs.biocuckoo.org/online.php (accessed on 20 February 2021)).

### 4.2. Analysis of the Characteristics of PtrARF Gene Family

*PtrARF* genes’ physical and chemical parameters, theoretical isoelectric point (pI), and molecular weight (Mw) were computed by ProtParam (https://web.expasy.org/protparam/ (accessed on 3 March 2021)). The subcellular localization sites were predicted by Plant-mPLoc (http://www.csbio.sjtu.edu.cn/bioinf/plant-multi/ (accessed on 3 March 2021)).

The gene structure map was drawn by the gene structure visualization server GSDS2.0 (http://gsds.gao-lab.org/ (accessed on 5 March 2021)). Based on the *Pfam Database*, each PtrARF protein domain was mapped by GSDS2.0. PtrARF protein sequences were submitted to MEME Suite (https://meme-suite.org/meme/tools/meme (accessed on 8 March 2021)) to predict the conserved motifs with default settings for parameters (the maximum number of motifs set to 20). The results were visualized by TBtools [60].

The conserved B3 domain (PF02362) sequences of *A. thaliana* [15], *O. sativa* [9], and *E. grandis* [48] were downloaded from the *PlantTFDB Database* (planttfdb.gao-lab.org (accessed on 15 March 2021)) and, together with the identified PtrARF proteins, were used for phylogenetic analysis, which was carried out according to the neighbor-joining (NJ) method using the ClustalX 2.0 and MEGA 5.2 software with a bootstrap of 1000 [61].

The non-synonymous (ka)/synonymous (ks) substitutions of *PtrARF* gene pairs in *P. trichocarpa* were identified by TBtools [60]. The synteny relationship of *PtrARF* genes between poplar and other three specie genomes was analyzed by TBtools and displayed by Dual Synteny Plotter [60].

### 4.3. Analysis of the Expression Characteristics of PtrARF Genes during Wood Formation

The expression values of *PtrARF* genes were obtained from our previous RNA-seq profiles of diverse development stems of poplar [44] and displayed by the heat maps generated by TBtools. In addition, to further uncover the accurate functions of *PtrARF* genes during the diverse processes of wood formation, the AspWood RNAseq dataset (https://popgenie.org/aspwood (accessed on 20 March 2021)) was also adopted to analyze the expression patterns of these *PtrARF* genes.

### 4.4. Plant Materials and Sample Collections

The plantlets of *P. trichocarpa* clone Nisqually^−1^ were planted in humus soil and grown under 16 h/8 h day/night photoperiod at 23–25 °C in the greenhouse at Northeast Forestry University for 90 days and then used as materials for reverse transcription quantitative real-time PCR (qRT-PCR) analysis. The samples were collected according to previous research [44] and stored at −80 °C for verifying the results of the expression profiles.

The plantlets of *P. trichocarpa* were cultured on a woody plant medium (WPM; pH 5.8) in the tissue culture laboratory under 16 h/8 h day/night photoperiod with light intensity of 50 µmol photons m^−2^ s^−1^ at 23–25 °C. After 3 weeks, seedlings were used for 100 μM ABA and 100 μM IAA treatment, respectively. The whole plants were sprayed in the super clean workbench until the leaves were about to drip. Each phytohormone treatment was applied for 0, 6, 12, 24, 48, and 72 h. The control group was treated with water at 0 h. Fresh roots, stems, and leaves from seedlings in each phytohormone treatment were sampled at the corresponding time points, immediately frozen in liquid nitrogen, and stored at −80 °C for analysis expression patterns of *PtrARF* genes in response to exogenous ABA and IAA treatment. Each sample consisted of three biological replicates.

### 4.5. qRT-PCR Analysis

The total RNA of samples was extracted using TaKaRa MiniBEST Plant RNA Extraction Kit (Takara, Dalian, China). The first cDNA strand was synthesized using PrimeScript™ RT reagent Kit with gDNA Eraser (Takara, Dalian, China).

To verify the results obtained from our previous expression profiles, 15 genes were selected to valid their expression levels in multiple stem segments of poplar using qRT-PCR. Primers sequences are listed in Appendix A. *PtrActin* was used as an internal reference gene [62]. The relative template abundance in each PCR expansion mixture was calculated by the 2^−ΔΔCT^ method [63]. Three biological replicates were used for gene expression analysis, and the expressions of the primary stem were set to 1 for normalization.

To test the response of *PtrARF* genes to exogenous ABA or IAA treatments, the qRT-PCR was performed and the primer sequences of *PtrARF* genes are listed in Appendix A. *PtrActin* was used as an internal reference gene [62]. The relative template abundance in each PCR expansion mixture was calculated by the 2^−ΔΔCT^ method [63]. Three biological replicates were used for gene expression analysis, and the expressions of the control samples (0 h) were set to 1 for normalization.

### 4.6. Determination of Subcellular Localization of PtrARF Proteins

The CDS of *PtrARF18*, *PtrARF23*, and *PtrARF29* without termination codon were amplified using specific primers (Appendix A) and then fused to the N-terminal of GFP under the control of CaMV 35S promoter in the pBI121 vector using kit (In-Fusion^®^ HD Cloning Kit, Takara, Dalian, China), respectively. The two fusion constructs were delivered into *P. trichocarpa* mesophyll protoplasts [64,65]. After incubation for 12–14 h, The GFP fluorescent images were photographed with confocal laser scanning microscope (Zeiss, Jena, Germany, LSM 800).

### 4.7. Yeast Two-Hybrid Assay

Based on the predicted interactions of *Arabidopsis* proteins by STRING (https://version11.string-db.org/cgi/input.pl?sessionId=8OEYdyOWWifa&input_page_show_search=on (accessed on 25 May 2021)), the ortholog pairs of AtARFs and AtIAAs in *P. trichocarpa* were used to carry out the ARF-IAA interaction analyses in poplar using the Y2H assay. The CDS of *PtrARF18*, *PtrARF23*, and *PtrARF29* were separately cloned into the pGBKT7 vector as bait, and the CDS of *PtrIAA10* and *PtrIAA28* were separately cloned into the pGADT7 vector as prey. First, pGBKT7-PtrARF18, pGBKT7-PtrARF23, and pGBKT7-PtrARF29 were transformed into the yeast strain and then spread into synthetically defined (SD) lacking Trp solid media at 30 °C for 3–5 d, while pGBKT7 vector was transformed as negative control. Transformed yeast cells were inoculated into SD/-Trp/-His/-Ade/X-α-Gal to test the autoactivation of these three PtrARFs. On the other hand, four various combinations of bait and prey vectors, i.e., pGBKT7-PtrARF23/pGADT7-PtrIAA28, pGBKT7-PtrARF23/pGADT7-PtrIAA10, pGBKT7-PtrARF29/pGADT7-PtrIAA28, and pGBKT7-PtrARF29/pGADT7-PtrIAA10, were co-transformed into the Y2H-Gold yeast strain. The pGADT7-T/pGBKT7-p53 pair and the pGADT7-T/pGBKT7-Lam pair were used as positive and negative control, respectively. After growth on SD/-Leu/-Trp medium for 3–5 d at 30 °C, the clones were transferred into the selective medium (SD/-Trp/-His/-Ade/X-α-Gal/AbA) at 30 °C for 3–5 d to test interactions.

### 4.8. Bimolecular Fluorescence Complementation (BiFC) Assay

BiFC assays were performed as described by Kerppola [66]. The coding regions of *PtrARF23* and *PtrARF29* without termination codon were amplified and ligated into pUC-nEYFP to produce PtrARF23: YFP^N^ and PtrARF29: YFP^N^, respectively. The coding regions of *PtrIAA10* and *PtrIAA28* were amplified by PCR and ligated into pUC-cEYFP after the digestion to produce PtrIAA10: YFP^C^ and PtrIAA28: YFP^C^, respectively. Combinations of the indicated plasmids were co-transformed into *P. trichocarpa* mesophyll protoplasts via PEG-calcium-mediated transformation [65]. YFP fluorescence was visualized 12–16 h after transformation under a confocal laser-scanning microscope (Zeiss, LSM 800).

## 5. Conclusions

In this study, a total of 35 *PtrARF* genes were identified based on *P. trichocarpa* v3.0, among which 16 *PtrARF* genes were preferentially expressed in primary stems, presumably showing a function of primary growth. In addition, 15 *PtrARF* genes were found to be significantly up-regulated in secondary stems. Their promoters contain at least one *cis*-element of SNBE, M46RE, ACI, ACII, ACIII, or SMRE, which play a role in SCW formation. Moreover, 15 *PtrARF* genes whose promoters contain at least one *cis*-element related to stress response and ABA treatments and 12 *PtrARF* genes whose promoters contain at least one *cis*-element related to response to IAA responded differently to exogenous ABA and IAA treatments, respectively. Furthermore, PtrARF18 was identified to be a transcriptional activator that could activate itself, too, while PtrARF23 and PtrARF29 were two repressors. Finally, we demonstrated that PtrARF23 interacted with PtrIAA10 and PtrIAA28, and PtrARF29 interacted with PtrIAA28.

## Figures and Tables

**Figure 1 ijms-24-00740-f001:**
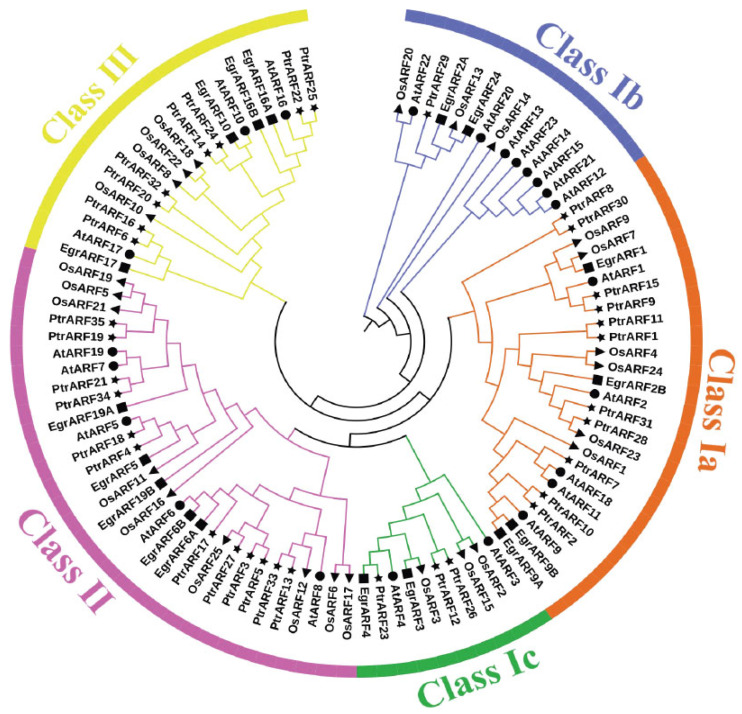
Phylogenetic analysis of the ARF proteins from *P. trichocarpa*, *A. thaliana*, *E. grandis*, and *O. sativa*. The conserved domain (PF02362) sequences of 35 PtrARF proteins, 23 AtARF proteins, 17 EgrARF proteins, and 25 OsARF proteins were aligned with Clustal X 2.0; the phylogenetic trees were constructed with MEGA 5.2 using the neighbor-joining (NJ) method and 1000 repetitions of bootstrap tests. All *ARF* genes were classified into three classes with different colors. Four species were represented by four symbols, respectively. Star, *P. trichocarpa*; round, *A. thaliana*; square, *E. grandis*; triangle, *O. sativa*.

**Figure 2 ijms-24-00740-f002:**
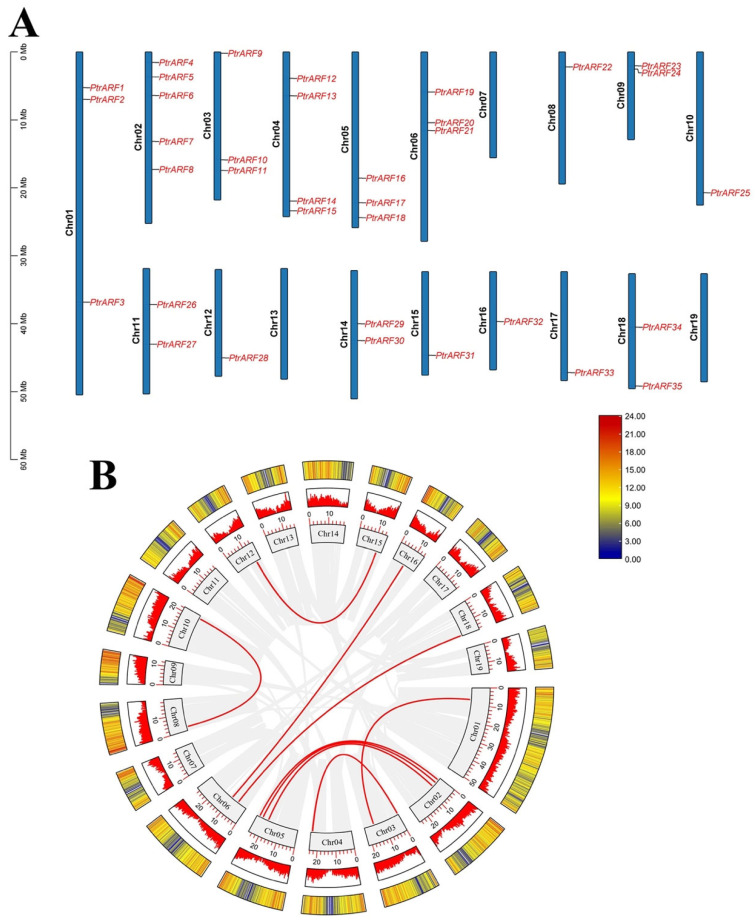
Chromosome distribution and synteny relationships of *PtrARF* gene family. (**A**) Chromosomal locations and duplicated gene pairs of distribution of 35 *PtrARF* genes. Each gene was mapped to the chromosome based on its physical location. The chromosome number (Chr01–Chr19) is indicated at the left; (**B**) circle map of the duplication gene pairs of the *PtrARF* genes. The heatmap and the histograms in rectangles represent the gene density on the chromosomes. The red lines represent collinear pairs of *PtrARF* genes, while the gray lines indicate collinear blocks of all poplar genes.

**Figure 3 ijms-24-00740-f003:**
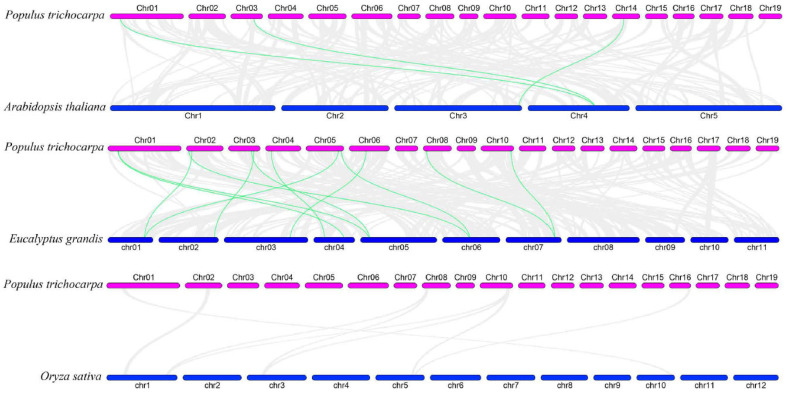
Synteny relationships of the *ARF* genes between poplar and three other plant species. The green lines highlight the syntenic *ARF* gene pairs, and the gray lines represent the collinear blocks in poplar that are orthologous to the other plant genomes.

**Figure 4 ijms-24-00740-f004:**
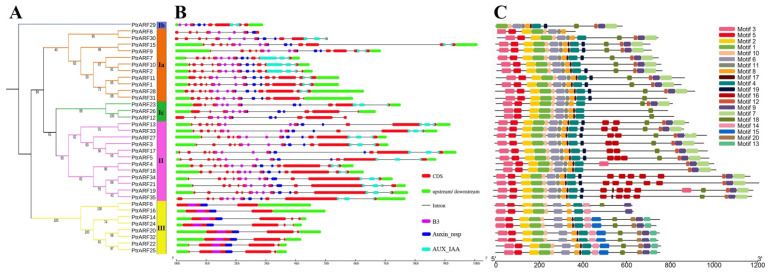
Phylogenetic tree, gene structures, and protein motifs of 35 *PtrARF* genes. (**A**) The phylogenetic tree. PtrARF protein sequences were aligned with Clustal X 2.0, and the phylogenetic tree was constructed using the neighbor-joining method; (**B**) Gene structures. Red boxes represent CDS regions, green boxes represent 5′UTR or 3′UTR regions, and black lines represent introns regions. The BD domains are indicated by purple boxes, the Auxin_resp domains are indicated by blue boxes, and the AUX_IAA domains are indicated by cyan boxes. The sizes of the exons and introns are estimated using the scale at the bottom; (**C**) protein motifs. The motif logos were drawn using Tbtools. Conserved motifs (1–20) are represented by different colored boxes, while non-conserved sequences are shown by black lines.

**Figure 5 ijms-24-00740-f005:**
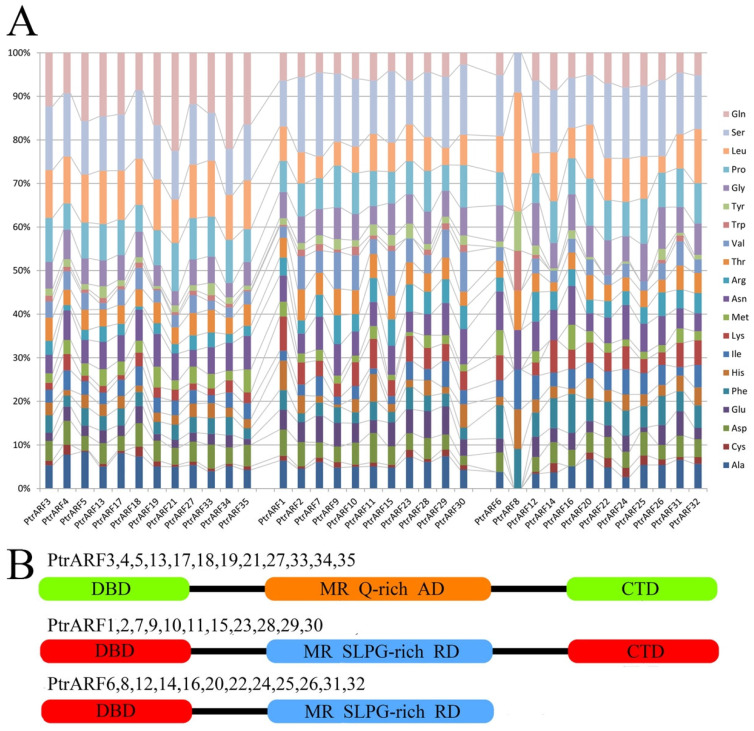
Amino acid compositions of middle region (MR) and classification of the PtrARF proteins. (**A**) Amino acid contents of the MR of 35 PtrARF proteins. PtrARF proteins are represented in the horizontal axis, and corresponding amino acids are represented in the vertical axis. Colored bars represent different amino acids; (**B**) the protein structure of PtrARF proteins. DBD, DNA-binding domain; CTD, C-terminal dimerization domain; MR, middle region; RD, repression domain; AD, activation domain; Q, glutamine; S, serine; L, leucine; P, proline; G, glycine.

**Figure 6 ijms-24-00740-f006:**
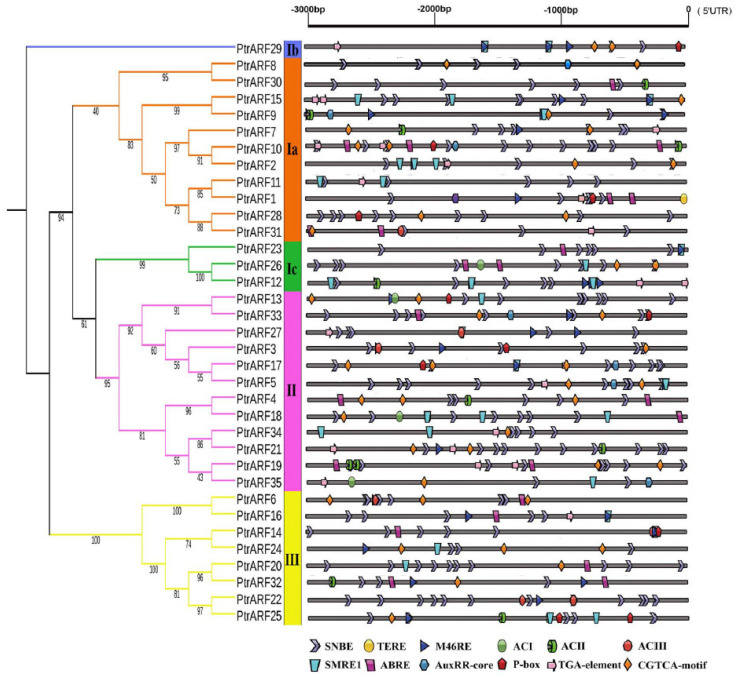
The *cis*-elements related to hormone response and xylem development in the promoters of 35 *PtrARF* genes. The 3000 bp promoter region (upstream DNA sequence of the 5′UTR) of each *PtrARF* gene was analyzed for *cis*-elements. Different *cis*-elements are indicated by different diagrams.

**Figure 7 ijms-24-00740-f007:**
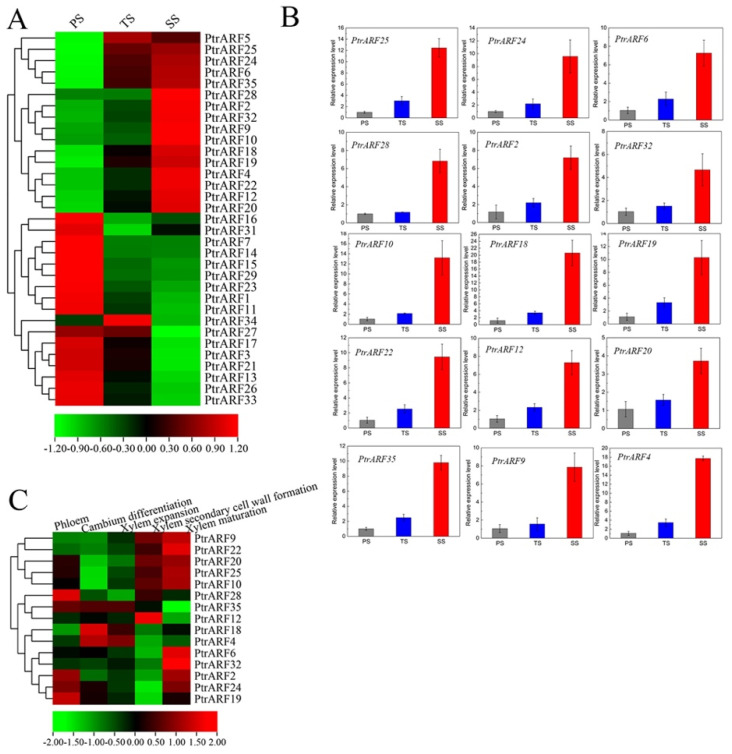
The expression patterns of *PtrARF* genes during wood formation. (**A**) Expression patterns of 35 *PtrARF* genes in primary stems (PS), transitional stems (TS), and secondary stems (SS) tissues in RNA-seq; (**B**) the expression level of 15 *PtrARF* genes in PS, TS, and SS of poplar using qRT-PCR assay. The expression levels of each gene were calculated in relevance to corresponding gene expression PS. Error bars represent standard deviation of biologic replicates. (**C**) The expression patterns of 15 *PtrARF* genes were obtained from AspWood RNAseq dataset during different phases of wood formation.

**Figure 8 ijms-24-00740-f008:**
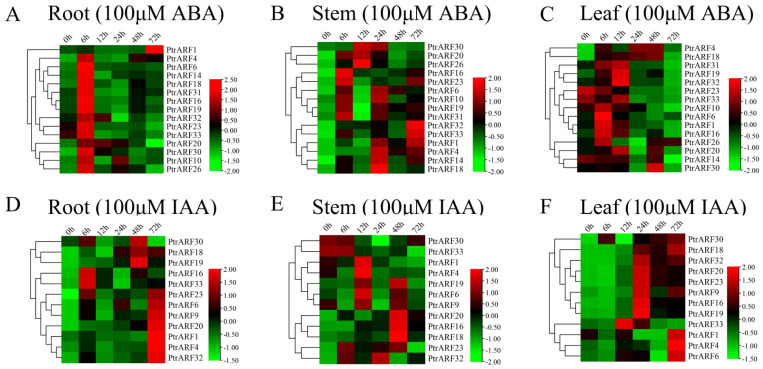
Spatio-temporal expression patterns of *PtrARF* genes under exogenous ABA and IAA treatment. (**A**–**C**) Heatmaps of relative gene expression levels treated with 100 μM ABA. (**D**–**F**) Heatmaps of relative gene expression levels treated with 100 μM IAA. The expression levels of each gene were calculated relative to its expression level at 0 h. We standardized each gene in heatmaps.

**Figure 9 ijms-24-00740-f009:**
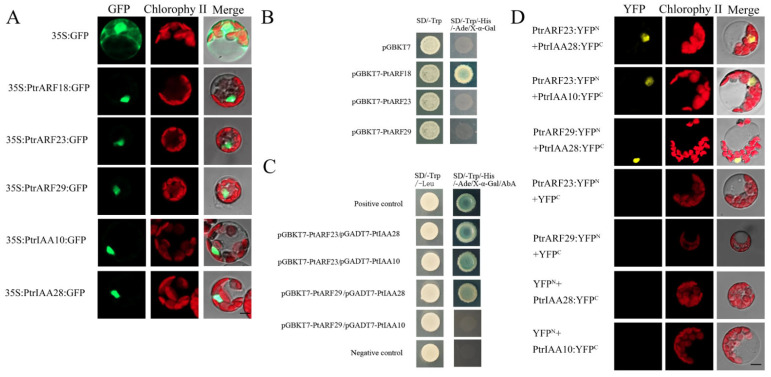
Subcellular localization, transcriptional self-activation, and protein–protein interactions. (**A**) Subcellular localization of PtrARF18, PtrARF23, PtrARF29, PtrIAA10, and PtrIAA28 was examined in poplar leaf protoplasts expressing green fluorescent protein (GFP); (**B**) transcriptional self-activation of PtrARF18, PtrARF23, and PtrARF29. The pGBKT7 vector-transformed yeast cells were used as negative control; (**C**) yeast two-hybrid assay of PtrARF23/PtrARF29 and PtrIAA10/PtrIAA28 interactions. pGBKT7-Lam/pGADT7-T- and pGBKT7–53/pGADT7-T-co-transformed yeast cells were used as negative and positive control, respectively; (**D**) bimolecular fluorescence complementation (BiFC) assay of PtrARF23/PtrARF29 and PtrIAA10/PtrIAA28 interactions. BiFC vectors of interaction proteins were co-transfected into *Populus* mesophyll protoplasts. Co-transfection of each protein of interest with empty plasmid was performed as a negative control. Scale bars, 10 μm.

## Data Availability

All data generated or analyzed during this study are included in this published article and information files. Informed consent was obtained from all subjects involved in the study. The raw sequencing data used during this study have been deposited in the NCBI’s SRA with the accession number PRJNA628501 (https://www.ncbi.nlm.nih.gov/bioproject/PRJNA628501 (accessed on 10 May 2021).

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
