# Peer review of "Genome-Wide Identification and Characterization of Auxin Response Factor (ARF) Gene Family Involved in Wood Formation and Response to Exogenous Hormone Treatment in Populus trichocarpa"

_ijms, 2023, doi:10.3390/ijms24010740_

Round 1
Reviewer 1 Report
The authors of the manuscript “Genome-Wide identification and characterization of Auxin Response Factor (ARF) gene family involved in wood formation and response to exogenous hormone treatment in Populus trichocarpa” have attempted and performed an illustrious work, important to the scientific world in the field of auxins response factors. There are some small issues where the author can concentrate while revising the manuscript prior to its final acceptance.
1. 39 ARFs were identified based on P. trichocarpa v1.1 database, while 35 in v 3.0. What is the difference?
2. In most previous studies, the full-length ORFs were used in phylogenetic analysis (including Arabidopsis), however, the typic conserved domain were used in this work. Is there any difference in phylogenetic classification of ARFs between using full-length ORFs and typic conserved domain?
3. Line 120-121, “ARF gene evolution was later than that of herbs and woody plant and monocotyledons and dicotyledons respectively” I think the phylogenetic tree was not enough to get this conclusion, can you explain?
4. Figure 2, 5, 7 should be improved by replaced by a clearer image.
5. Line 370, why chose ARF18/23/29 for further analysis, the reason is ?
6. I think the author should give the ID of PtrIAA10 and 28.
7. Disscussion
There were 39 PtrARFs have been identified in previous work, what is the difference and improvement of your work? I think this is an important part in discussion.
8. Line 19, 24, 259, 260, 263…… cis should be in italic
Reviewer 2 Report
The original research manuscript entitled,” Genome-Wide identification and characterization of ARF gene family involved in wood formation and response to exogenous hormone treatment in Populus trichocarpa” is a comprehensive study of important Auxin-responsive genes family. The authors have thoroughly identified, analyzed, and performed some basic experiments to reveal the possible role of genes in various functions. The manuscript presents a good overview of the selected gene family and a well-structured analysis was performed; however, I suggest a major modification before further processing of the article.
The authors are recommended to revise the manuscript according to the following suggestions
1) Line 16- is incomplete and not mentions about what kind of processes, please revise the sentence.
2) Line 20 “investigated and analyzed” simply use anyone and use short sentences.
3) Provide enough reason to conduct this study since this gene family is already studied in the same/similar plant species
4) Figure-2 genes are shown in Chromosomes so the names should be in italics
5) More protein properties like pI, GRAVY, and predicted 3D structure can be added to the supplementary file
6) Make sure to correct gene name and protein names in correct format.
7) The Motif details should be added to the supplementary file
8) Section 2.5 can be further improved
9) Figure 7 make sure to write gene names in the correct manner
10) Section 3.2 should be discussed in a more precise way. The discussion is not satisfactory.
Reviewer 3 Report
The best work I ever have reviewed!
- The article is original and important considering auxin signaling in the woody plants. Auxin is one of the most important plant hormone with complicated regulating ways. ARFs play crucial roles in this ways and most studies of them were made on Arabidopsis and, later, other herbaceous plants. This work made impressive in-depth study of specific features of ARFs in model woody plant Populus trichocarpa.
- From my point of view all parts of article are very clear and full, easy to understand, I see no sections need more explanation, but other readers could have other opinions for sure.
- The study identified all 35 PtARF genes P.trychocarpa v3.0, investigated their preferential expression points, dependence on IAA and ABA – expected according to promoter structure and appearing after hormone treatment, validated the interactions of three of PtrARFs with specific inhibitors of auxin signaling.
- All parts of research demonstrate clear and accurate methodology, corresponding use of modern programs and techniques, adequate analysis of data.
- Conclusions does correspond to the arguments and illustrations, and address well the main question.
- All the reference I have trace were appropriate and accurate.
- In the printed version of article there could be a bit difficult to read the names of genes at Fig.1, 2, and especially 4, and numbers at the fig.2(scales), 5, 6 and especially 4. In the web version it make no problem to enlarge the picture. If it possible, for printing I’d offer turn this Figures and give the whole page for each of them.
- It seems a bit complicated wording in the last two phrases of Conclusion “. Moreover, PtrARF18 was identified to be a transcriptional self-activator activator localized in nucleus, while PtrARF23 and PtrARF29 were repressors localized in nucleus. Furthermore, we validated that PtrARF23 interacted with PtrIAA10 and PtrI-AA28 in nucleus, while PtrARF29 interacted with PtIAA28 in nucleus.” From my point of view it’s enough to say once all this proteins are localized in nucleus and make accents on differences of their functions and interactions.
